

# Supervised deep learning embeddings for the prediction of cervical cancer diagnosis

Kelwin Fernandes[1,2], Davide Chicco[3], Jaime S. Cardoso[1,2] and Jessica Fernandes[4]

[1] Instituto de Engenharia de Sistemas e Computadores Tecnologia e Ciencia (INESC TEC), Porto, Portugal
[2] Universidade do Porto, Porto, Portugal
[3] Princess Margaret Cancer Centre, Toronto, ON, Canada
[4] Universidad Central de Venezuela, Caracas, Venezuela

## ABSTRACT

Cervical cancer remains a significant cause of mortality all around the world, even if it can be prevented and cured by removing affected tissues in early stages. Providing universal and efficient access to cervical screening programs is a challenge that requires identifying vulnerable individuals in the population, among other steps. In this work, we present a computationally automated strategy for predicting the outcome of the patient biopsy, given risk patterns from individual medical records. We propose a machine learning technique that allows a joint and fully supervised optimization of dimensionality reduction and classification models. We also build a model able to highlight relevant properties in the low dimensional space, to ease the classification of patients. We instantiated the proposed approach with deep learning architectures, and achieved accurate prediction results (top area under the curve AUC = 0.6875) which outperform previously developed methods, such as denoising autoencoders. Additionally, we explored some clinical findings from the embedding spaces, and we validated them through the medical literature, making them reliable for physicians and biomedical researchers.

## INTRODUCTION

Despite the possibility of prevention with regular cytological screening, cervical cancer remains a significant cause of mortality in low-income countries (*Kauffman et al., 2013*). The cervical tumor is the cause of more than 500,000 cases per year, and kills more than 250,000 patients in the same period, worldwide (*Fernandes, Cardoso & Fernandes, 2015*). However, cervical cancer can be prevented by means of the human papillomavirus infection (HPV) vaccine, and regular low-cost screening programs (*Centers for Disease Control and Prevention (CDC), 2013*). The two most widespread techniques in screening programs are conventional or liquid cytology and colposcopy (*Fernandes, Cardoso & Fernandes, 2015*; *Plissiti & Nikou, 2013*; *Fernandes, Cardoso & Fernandes, 2017b*; *Xu et al., 2016*). Furthermore, this cancer can be cured by removing the affected tissues when

Corresponding author
Kelwin Fernandes, kafc@inesctec.pt

identified in early stages (*Fernandes, Cardoso & Fernandes, 2015*; *Centers for Disease Control and Prevention (CDC), 2013*), in most cases.

The development of cervical cancer is usually slow and preceded by abnormalities in the cervix (dysplasia). However, the absence of early stage symptoms might cause carelessness in prevention. Additionally, in developing countries, there is a lack of resources, and patients usually have poor adherence to routine screening due to low problem awareness.

While improving the resection of lesions in the first visits has a direct impact on patients that attend screening programs, the most vulnerable populations have poor or even non-existent adherence to treatment programs. Scarce awareness of the problem and patients' discomfort with the medical procedure might be the main causes of this problem. Furthermore, in low-income countries, this issue can be due to lack of access to vulnerable populations with low access to information and medical centers. Consequently, the computational prediction of individual patient risk has a key role in this context. Identifying patients with the highest risk of developing cervical cancer can improve the targeting efficacy of cervical cancer screening programs: our software performs this operation computationally in a few minutes by producing accurate prediction scores.

*Fernandes, Cardoso & Fernandes (2017b)* performed a preliminary attempt to tackle the problem of predicting the patient's risk to develop cervical cancer through machine learning software. In that project, the authors employed transfer learning strategies for the prediction of the individual patient risk on a dataset of cervical patient medical tests. They focused on transferring knowledge between linear classifiers on similar tasks, to predict the patient's risk (*Fernandes, Cardoso & Fernandes, 2017b*).

Given the high sparsity of the associated risk factors in the population, dimensionality reduction techniques can improve the robustness of the machine learning predictive models. However, many projects that take advantage of dimensionality reduction and classification use suboptimal approaches, where each component is learned separately (*Li et al., 2012*; *Bessa et al., 2014*; *Lacoste-Julien, Sha & Jordan, 2009*).

In this work, we propose a joint strategy to learn the low-dimensional space and the classifier itself in a fully supervised way. Our strategy is able to reduce class overlap by concentrating observations from the healthy patients class into a single point of the space, while retaining as much information as possible from the patients with high risk of developing cervical cancer.

We based our prediction algorithm on artificial neural networks (ANNs), which are machine learning methods able to discover non-linear patterns by means of aggregation of functions with non-linear activations. A recent trend in this field is deep learning (*LeCun, Bengio & Hinton, 2015*), which involves large neural network architectures with successive applications of such functions. Deep learning, in fact, has been able to provide accurate predictions of patient diagnosis in multiple medical domains (*Xu et al., 2016*; *Chicco, Sadowski & Baldi, 2014*; *Fernandes, Cardoso & Astrup, 2017a*; *Cangelosi et al., 2016*; *Alipanahi et al., 2015*). We applied our learning scheme to deep variational autoencoders and feed-forward neural networks. Finally, we explored

visualization techniques to understand and validate the medical concepts captured by the embeddings.

We organize the rest of the paper as follows. After this Introduction, we describe the proposed method and the dataset analyzed in the Methods and Dataset sections. Afterwards, we describe the computational prediction results in the Results section, the model outcome interpretation in the Discussion section, and we conclude the manuscript outlining some conclusion and future development.

## METHODS

High dimensional data can lead to several problems: in addition to high computational costs (in memory and time), it often leads to overfitting (*Van Der Maaten, Postma & Van den Herik, 2009*; *Chicco, 2017*; *Moore, 2004*). Dimensionality reduction can limit these problems and, additionally, can improve the visualization and interpretation of the dataset, because it allows researchers to focus on a reduced number of features. For these reasons, we decided to map the original dataset features into a reduced dimensionality before performing the classification task.

Generally, to tackle high-dimensional classification problems, machine learning traditional approaches attempt to reduce the high-dimensional feature space to a low-dimensional one, to facilitate the posterior fitting of a predictive model. In many cases, researchers perform these two steps separately, deriving suboptimal combined models (*Li et al., 2012*; *Bessa et al., 2014*; *Lacoste-Julien, Sha & Jordan, 2009*). Moreover, since dimensionality reduction techniques are often learned in an unsupervised fashion, they are unable to preserve and exploit the separability between observations from different classes.

In dimensionality reduction, researchers use two categories of objective functions: one for maximizing the model capability of recovering the original feature space from the compressed low dimensional one, and another one for maximizing the consistency of pairwise similarities in both high and low dimensional spaces.

Since defining a similarity metric in a high-dimensional space might be difficult, we limit the scope of this work to minimizing the reconstruction loss. In this sense, given a set of labeled input vectors $X = \{x_1, x_2, \ldots, x_n\}$, where $x_i \in \mathrm{R}^d, \forall i \in 1, \ldots, n$ and $Y$ is a vector with the labels associated to each observation, we want to obtain two functions $C: \mathrm{R}^d \to \mathrm{R}^m$ and $D: \mathrm{R}^m \to \mathrm{R}^d$ such that $m < d$ and that minimizes the following loss:

$$L_r(C, D, X) = \frac{1}{|X|} \sum_{x \in X} ((D \circ C)(x) - x)^2 \tag{1}$$

Namely, the composition ($\circ$) of the compressing ($C$), and decompressing ($D$) functions approximate the identity function.

In the following sections, we describe the proposed dimensionality reduction technique and its instantiation to deep learning architectures.

## Joint dimensionality reduction and classification

Since our final goal is to classify the data instances (observations), we need to achieve a good low-dimensional mapping and build the classifier independently. Thereby, we propose a joint loss function that minimizes the trade-off between data reconstruction and classification performance:

$$L(M, C, D, X, Y) = L_c((M \circ C)(X), Y) + \lambda L_r(C, D, X) \qquad (2)$$

where $M$ is a classifier that receives as input the vectors in the low dimensional space $(C(X))$, $L_c$ is a classification loss function such as categorical cross-entropy, and $\lambda \geq 0$. In this case, we focus on the classification performance using Eq. (1) as a regularization factor of the models of interest. Hereafter, we will denote this method as semi-supervised dimensionality reduction.

## Fully supervised embeddings

The previously proposed loss function consists of two components: a supervised component given by the classification task, and an unsupervised component given by the low-dimensional mapping. However, the scientific community aims at understanding the properties captured in the embeddings, especially on visual and text embeddings (*Kiros, Salakhutdinov & Zemel, 2014*; *Levy, Goldberg & Ramat-Gan, 2014*). Moreover, inducing properties in the low-dimensional space can improve the class separability. To apply this enhancement, we introduce partial supervision in the $L_r$ loss.

We can explore these properties by learning the dimensionality reduction process in a supervised way. Namely, learning a bottleneck supervised mapping function $((D \circ C)(x) \approx M(x, y))$ instead of the traditional identity function $((D \circ C)(x) \approx x)$ used in reconstruction-based dimensionality reduction techniques. The reconstruction loss $L_r(C, D, X)$ becomes:

$$L_M(C, D, X, Y) = \frac{1}{|X|} \sum_{\langle x,y \rangle \in X, Y} ((D \circ C)(x) - M(x, y))^2 \qquad (3)$$

where $M(x)$ is the desired supervised mapping.

To facilitate the classification task, removing the overlap between both classes should be captured in low-dimensional spaces. Without loss of generality, we assume that the feature space is non-negative. Thereby we favor models with high linear separability between observations by using the mapping function Eq. (4) in Eq. (3).

$$\text{Sym}(x, y) = \begin{cases} x, & \text{if } y \\ -x, & \text{if} \neg y \end{cases} \qquad (4)$$

In our application, if all the features are non-negative, the optimal patient's behavior associates to the zero vector with total lack of risk patterns. On the other hand, a patient with high feature values is prone to have cancer. Within the context of cervical cancer screening, we propose the mapping given by Eq. (5), where the decoded version of the healthy patients is the zero vector. This idea resembles the fact that their risk conduct

has not contributed to the disease occurrence. On the other hand, we mapped ill patients to their original feature space, for promoting the low-dimensional vectors to explain the original risk patterns that originated the disease.

$$\text{Zero}(x, y) = 1(y) \cdot x \qquad (5)$$

While the definition of the properties of interest to be captured by the low-dimensional space is application-dependent, the strategy to promote such behavior can be adapted to other contexts.

## Deep supervised autoencoders

Autoencoders are special cases of deep neural networks for dimensionality reduction (*Chicco, Sadowski & Baldi, 2014*; *Vincent et al., 2008*). They can be seen as general feed-forward neural networks with two main sub-components: the first part of the neural network is known as the *encoder*, and its main purpose is to compress the feature space. The neural network achieves this step by using hidden layers with fewer units than the input features, or by enforcing sparsity in the hidden representation. The second part of the neural network, also known as the *decoder*, behaves in the opposite way, and tries to approximate the inverse encoding function. While these two components correspond to the $C$ and $D$ functions in Eq. (1), respectively, they can be broadly seen as a single ANN that learns the identity function through a bottleneck, a low number of units, or through sparse activations. Autoencoders are usually learned in an unsupervised fashion by minimizing the quadratic error Eq. (1).

Denoising autoencoders (DA) represent a special case of deep autoencoders that attempt to reconstruct the input vector when given a corrupted version (*Vincent et al., 2008*). DA can learn valuable representations even in the presence of noise. Scientists can experiment this task by adding an artificial source of noise in the input vectors. In the neural network architecture (Fig. 1), we also included a dropout layer after the input layer that randomly turns off at maximum one feature per patient (*Srivastava et al., 2014*). Thereby, we aim to build stable classifiers that produce similar outcomes for patients with small differences in their historical records. Furthermore, we aim at producing stable decisions when patients lie on a subset of the answers to the doctors' questions during the medical visit, by indicating absence of a given risk behavior (for example, high number of sexual partners, drug consumption, and others). We use a Parametric Rectifier Linear Unit (PReLU) (*He et al., 2015*) as activation function in the hidden layers of our architectures (Fig. 1). PReLU is a generalization of standard rectifier activation units, which can improve model fitting with low additional computational cost (*He et al., 2015*).

The loss functions (Eqs. 2 and 3) can learn a joint classification and encoding–decoding network in a multitask fashion (Fig. 2). Additionally, to allow the neural network to use either the learned or the original representation, we include a *bypass layer* that concatenates the hidden representation with the corrupted input. In the past, researchers have used this technique in biomedical image segmentation with U-net architectures (*Ronneberger, Fischer & Brox, 2015*) to recover possible losses in the compression process,

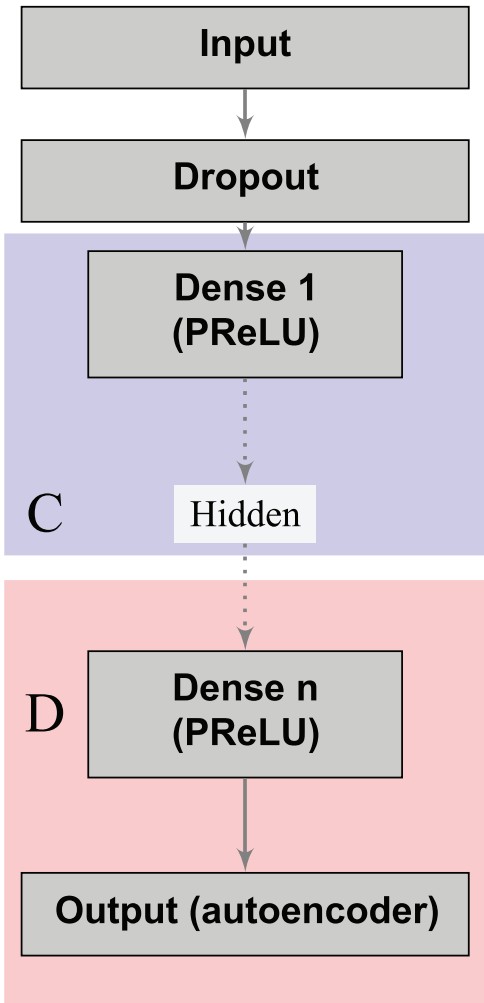

**Figure 1  Deep denoising autoencoder.** The blocks in blue and red represent the encoding (*C*) and decoding (*D*) components of the network, respectively.

and to reduce the problem of vanishing gradients. We use this *bypass layer* with cross-validation.

In a nutshell, our contribution can be summarized as follows: (i) we formalized a loss function to handle dimensionality reduction and classification in a joint fashion, leading to a global optimal pipeline; (ii) in order to induce desired properties on the compressed space, we proposed a loss that measures the model's capability to recreate a mapping with the desired property instead of the identity function usually applied in dimensionality reduction; (iii) we showed that multitask autoencoders based on neural networks can be used as a specific instance to solve this problem, and we instantiated this idea to model an individual patient's risk of having cervical cancer.

## DATASET

The dataset we analyze contains medical records of 858 patients, and covers a random sampling of patients between 2012 and 2013 who attended the gynecology service at

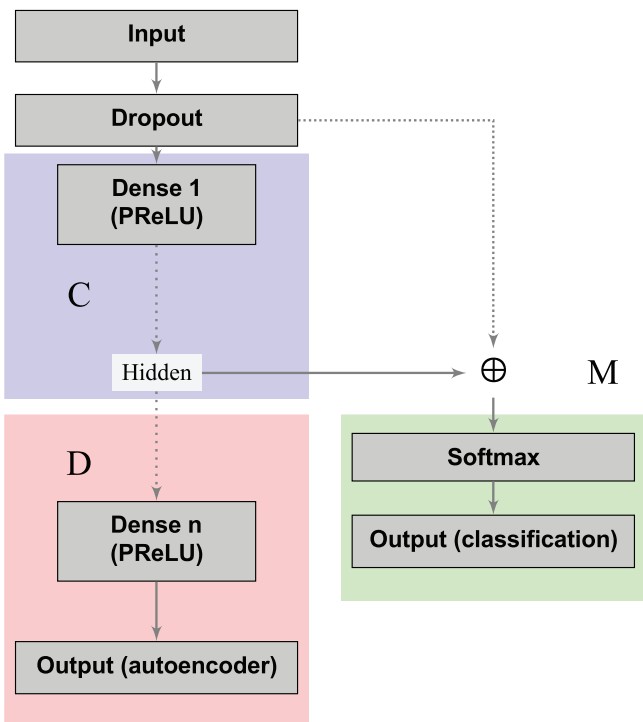

**Figure 2 Supervised deep embedding architecture.** The blocks in blue, red, and green represent the encoding (C), decoding (D), and classification (M) components of the network, respectively.

Hospital Universitario de Caracas in Caracas, Venezuela. Most of the patients belong to the lowest socioeconomic status (Graffar classification: IV–V (*Graffar, 1956*)) with low income and educational level, being the population with the highest risk. The age of the patients spans between 13 and 84 years old (27 years old on average). All patients are sexually active and most of them (98%) have been pregnant at least once. The screening process covers traditional cytology, the colposcopic assessment with acetic acid and the Schiller test (Lugol's iodine solution) (*Fernandes, Cardoso & Fernandes, 2017b*). The medical records include the age of the patient, sexual activity (number of sexual partners and age of first sexual intercourse), number of pregnancies, smoking behavior, use of contraceptives (hormonal and intrauterine devices), and historical records of sexually transmitted diseases (STDs) (Table 1). Hence, we encoded the features denoted by bool × $T$, $T \in$ {bool, int} as two independent values: whether or not the patient answered the question and, if she did, the answered value. In some cases, the patients decided not to answer some questions for privacy concerns. This behavior is often associated with risk behaviors being a relevant feature to explore when modeling risk patterns. Therefore, we added a flag feature that allows the model to identify if the question was answered or not after missing value imputation. We encoded the categorical features using the one-of-K scheme. The hospital anonymized all the records before releasing the dataset. The dataset is now publically available on the Machine Learning Repository website of the University of California Irvine (UCI ML) (*University of California Irvine, 1987*),

**Table 1 Feature names and data type acquired in the risk factors dataset (*Fernandes, Cardoso & Fernandes, 2017b*).**

| Feature | Type | Feature | Type |
|---|---|---|---|
| Age | int | IUD (years) | int |
| Number of sexual partners | bool × int | Sexually transmitted diseases (STDs) (yes/no) | bool × bool |
| Age of first sexual intercourse | bool × int | Number of STDs | int |
| Number of pregnancies | bool × int | Diagnosed STDs | Categorical |
| Smokes (yes/no) | bool × bool | STDs (years since first diagnosis) | int |
| Smokes (years and packs) | int × int | STDs (years last diagnosis) | int |
| Hormonal contraceptives (yes/no) | bool | Previous cervical diagnosis (yes/no) | bool |
| Hormonal contraceptives (years) | int | Previous cervical diagnosis (years) | int |
| Intrauterine device (IUD) (yes/no) | bool | Previous cervical diagnosis | Categorical |

**Note:**
int, integer; bool, boolean.

**Table 2 Set of possible options for fine-tuning each parameter.**

| Parameter | Values |
|---|---|
| Depth | {1, . . . , 6} |
| Width | {10, 20} |
| Regularization | {0.01, 0.1} |
| Bypass usage | {*false*, *true*} |

which also contains a description of the features (*University of California Irvine Machine Learning Repository, 2017*).

To avoid problems of the algorithm behavior related to different value ranges of each feature, we scaled all the features in our experiments using [0,1] normalization, and we input missing data using the average value (*Chicco, 2017*). While more complex pre-processing schemes could be introduced, such as inferring the missing value with a *k*-nearest neighbor model (*Santos et al., 2015*), we decided to use this methodology to avoid additional complexity that would make it difficult to fairly compare the explored techniques. In most cases, the features positively correlate to the cancer variable, with 0 representing the lack of that risk pattern and 1 representing the maximum risk.

## RESULTS

We measured the performance of the proposed methods with the area under the *Precision–Recall* (PR) curves (*Davis & Goadrich, 2006*; *Chicco, 2017*) and the *logistic loss* (also known as *cross-entropy loss*) function.

As baseline, we use a deep feed-forward neural network with a softmax activation in the output layer. The remaining parameters (such as the initial dropout layer, depth and optimization algorithm) conform to the ones used in the proposed methodologies (Table 2). The main hyper-parameters related to the network topology are the depth and width, which define the number of layers in the architecture and the size of the low-dimensional representation.

**Table 3 Performance of the proposed architectures in terms of area under the Precision–Recall curve.**

| Subset | Baseline | Semi | Sym | Zero | SVM | k-NN | DecTree |
|---|---|---|---|---|---|---|---|
|  | 0.1334 | 0.1424 | 0.1534 | 0.1744 | 0.0877 | 0.0345 | **0.1941** |
| C | 0.1998 | 0.1853 | 0.2115 | 0.2174 | 0.1550 | **0.3033** | 0.2560 |
| H | 0.4536 | 0.4459 | 0.4407 | **0.4625** | 0.4192 | 0.3885 | 0.3616 |
| S | 0.6416 | 0.6411 | 0.6335 | **0.6668** | 0.5905 | 0.5681 | 0.6242 |
| CH | 0.4752 | 0.4684 | **0.4754** | 0.4609 | 0.4423 | 0.4095 | 0.4023 |
| CS | 0.6265 | **0.6424** | 0.6388 | 0.5985 | 0.6205 | 0.5379 | 0.6089 |
| HS | 0.6200 | **0.6356** | 0.6277 | 0.5864 | 0.6199 | 0.6335 | 0.5956 |
| CHS | 0.6665 | 0.6351 | **0.6875** | 0.6404 | 0.6374 | 0.6653 | 0.5542 |
| **Best** | 0 | 2 | 2 | 2 | 0 | 1 | 1 |

Notes:
The subset of observable screening strategies include: Cytology (C), Hinselmann (H), and Schiller (S).
Baseline, deep feed-forward neural network; Semi, semi-supervised dimensionality reduction (Eq. 2); Sym, symmetry mapping dimensionality reduction (Eq. 4); Zero, zero mapping dimensionality reduction (Eq. 5); SVM, support vector machine; k-NN, k-nearest neighbors; DecTree, decision tree.
We highlight the best performing models in bold.

We used a stratified 10-fold cross-validation in the assessment of the proposed methods. We optimized the neural networks by using the RMSProp optimization strategy (*Tieleman & Hinton, 2012*) for a maximum number of 500 epochs, with early stopping after 100 iterations without improvement and a batch size of 32. We validated these parameters empirically, and it was enough to ensure model convergence in all cases. We also validated the performance of other optimization strategies such as *Adam* and stochastic gradient descent. However, we did not observe any gain in terms of predictive performance or convergence. We use sparse autoencoders by adding an $L_1$ penalization term, to ensure that each unit combines a small subset of risk factors, as would be done by a human expert.

We fine-tuned all the hyper-parameters using a grid search strategy with nested stratified threefold cross-validation. In this sense, we validated the performance of each network configuration on three training-validation partitions and choose the one that maximizes the area under the PR curve. Then, for the best configuration, we re-trained the model using the entire training set. We chose the size of the low-dimensional space as part of this nested cross-validation procedure, and chose empirically the parameters related to the optimization algorithm (that are strategy, number of epochs, early stopping).

To recreate the decisions made by the physician at different configurations of the screening process, we consider the observability of all possible subsets of screening outcomes when predicting the biopsy results. Thereby, we cover scenarios where only behavioral and demographic information is observable (first line of each table with empty subset) up to settings where cytology and colposcopy (Hinselmann and Schiller) results are available.

### Diagnosis prediction results

Our proposed architectures with embedding regularization achieved the best diagnosis prediction results in most cases (Tables 3 and 4) when compared with other neural

**Table 4 Performance of the proposed architectures in terms of logarithmic loss.**

| Subset | Baseline | Semi | Sym | Zero | SVM | k-NN | DecTree |
|--------|----------|------|------|------|-----|------|---------|
|        | 0.3004   | 0.2708 | 0.2657 | 0.2716 | **0.2421** | 4.3670 | 4.1889 |
| C      | 0.2829   | 0.2757 | 0.2868 | **0.2609** | 0.2614 | 2.6884 | 3.5001 |
| H      | 0.2169   | 0.2274 | 0.2422 | 0.2031 | 0.1984 | 0.7178 | 3.2175 |
| S      | 0.1710   | 0.1475 | 0.1489 | 0.1359 | **0.1273** | 0.9366 | 1.6893 |
| CH     | 0.2210   | **0.2054** | 0.2286 | 0.2123 | 0.2196 | 1.0477 | 2.8509 |
| CS     | 0.1594   | 0.1469 | **0.1240** | 0.1464 | 0.1248 | 0.4036 | 1.7687 |
| HS     | 0.1632   | 0.1786 | 0.1615 | 0.1622 | **0.1225** | 0.3238 | 1.8098 |
| CHS    | 0.1563   | 0.1577 | 0.1494 | 0.1514 | **0.1099** | 0.4037 | 1.8906 |
| **Best** | 0      | 1    | 1    | 1    | 4   | 0    | 0       |

Notes:
The subset of observable screening strategies include: Cytology (C), Hinselmann (H), and Schiller (S). Area under the Precision–Recall curve.
Baseline, deep feed-forward neural network; Semi, semi-supervised dimensionality reduction (Eq. 2); Sym, symmetry mapping dimensionality reduction (Eq. 4); Zero, zero mapping dimensionality reduction (Eq. 5); SVM, support vector machine; k-NN, k-nearest neighbors; DecTree, decision tree.
We highlight the best performing models in bold.

network approaches. Furthermore, the fully supervised embeddings improved the performance of the semi-supervised approach (Eq. 2), through both the strategies (symmetric and zero mapping). The relative gains in terms of area under the PR curve depend on the subset of observable modalities, ranging from 30.7% when only medical records are observed to 3.3% when the outcome of all the screening procedures is known.

Using a paired difference Student's t-test (*Menke & Martinez, 2004*) with a 95% confidence level, zero-mapping methodology achieved better results than the baseline and semi-supervised learning schemes. We found no statistical differences between the symmetry and zero mappings.

We validated the performance of traditional machine learning models such as support vector machines (SVM) with radial basis function kernel (*Scholkopf et al., 1997*), k-nearest neighbors (*Peterson, 2009*), and decision trees (*Quinlan, 1986*). In general, the proposed models surpassed the performance of the classical methodologies in terms of area under the PR curve. The SVM model achieved better logarithmic loss given the post-processing of its scores using the Logistic Regression model that directly optimize this metric. Further improvements could be observed by post-processing the outcome of the other strategies.

The gains achieved by the mapping-based supervised embeddings happen because the proposed fully-supervised strategies aim to reduce the overlap between observations from both classes. In the past, researchers showed that class overlap has higher correlation with the model performance than the imbalance ratio in highly unbalanced datasets (*Cruz et al., 2016*). The visualization of the embeddings through the t-distributed stochastic neighbor embedding (t-SNE) (*Van Der Maaten & Hinton, 2008*) confirms this aspect, because in t-SNE fully supervised embeddings achieve better separability and fewer overlapping clusters (Figs. 3–6).

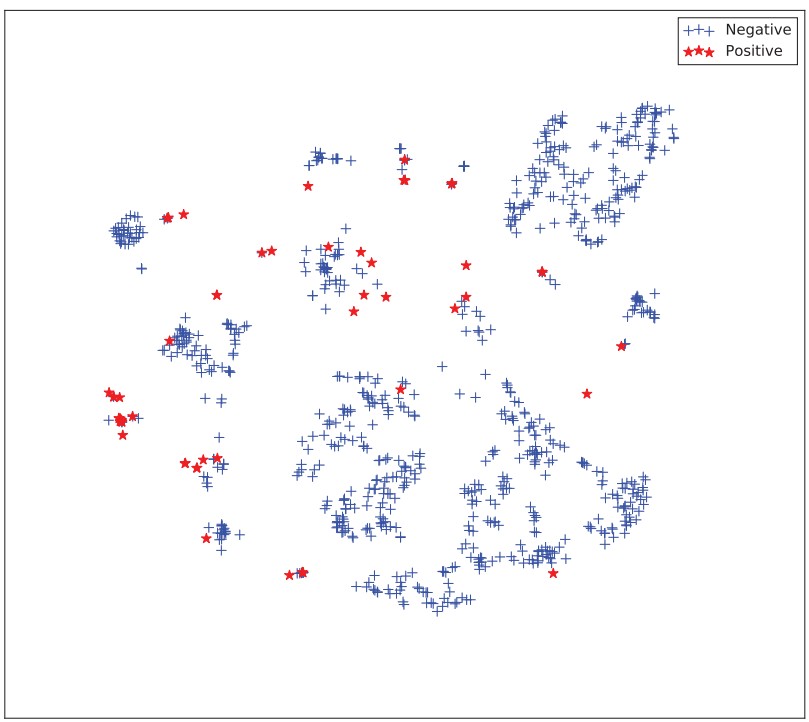

**Figure 3 Two-dimensional projection of the unsupervised embedding using t-distributed stochastic neighbor embedding (t-SNE) (*Van Der Maaten & Hinton, 2008*).**

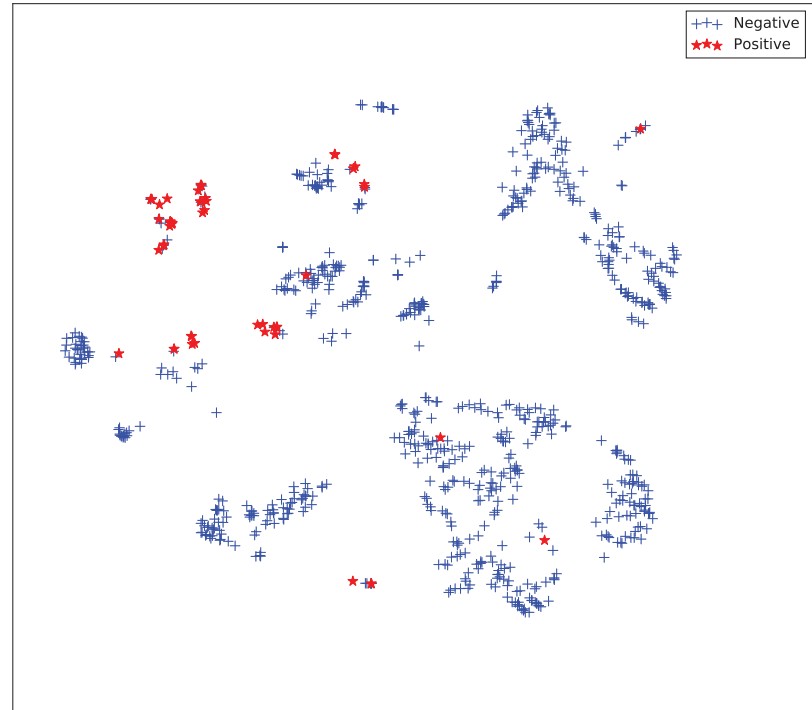

**Figure 4 Two-dimensional projection of the semi-supervised embedding using t-distributed stochastic neighbor embedding (t-SNE) (*Van Der Maaten & Hinton, 2008*).**

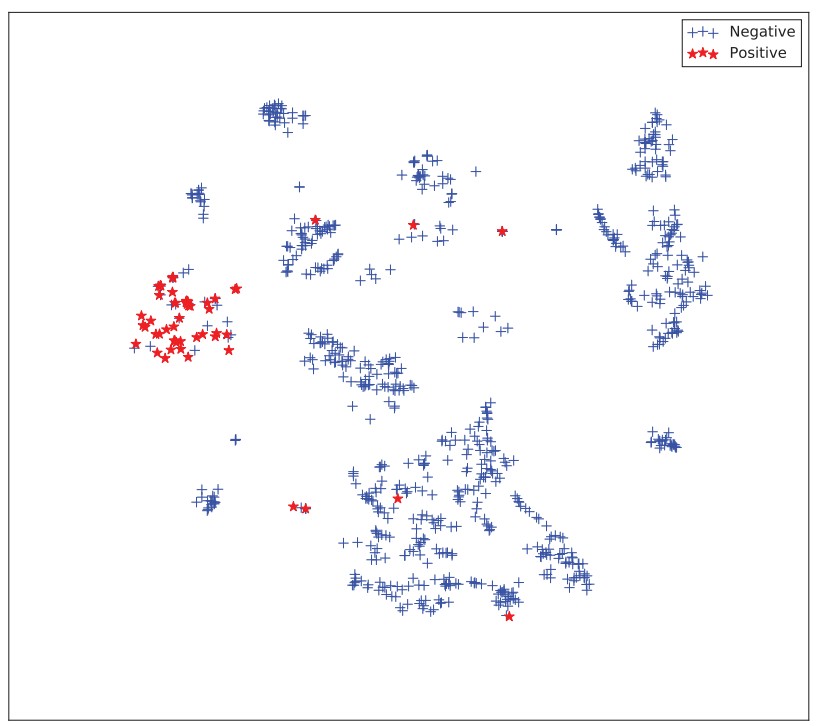

**Figure 5** Two-dimensional projection of the semi-supervised embedding with symmetry mapping using t-distributed stochastic neighbor embedding (t-SNE) (*Van Der Maaten & Hinton, 2008*).

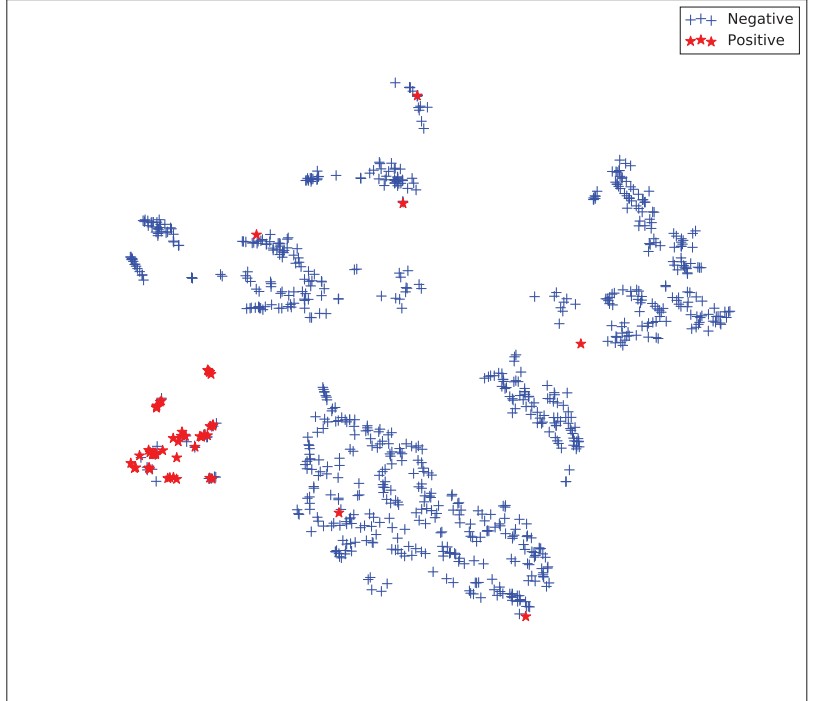

**Figure 6** Two-dimensional projection of the supervised embedding with zero mapping using t-distributed stochastic neighbor embedding (t-SNE) (*Van Der Maaten & Hinton, 2008*).

**Table 5 Performance of the proposed architectures on other datasets downloaded from UC Irvine Machine Learning Repository (*University of California Irvine, 1987*), measured through the area under the Precision–Recall curve.**

| Dataset | | Baseline | Semi | Sym | Zero |
|---|---|---|---|---|---|
| Breast cancer | *Mangasarian, Street & Wolberg (1995)* | 0.9795 | **0.9864** | 0.9835 | 0.9856 |
| Mammography | *Elter, Schulz-Wendtland & Wittenberg (2007)* | **0.8551** | 0.8539 | 0.8533 | 0.8489 |
| Parkinson | *Little et al. (2007)* | 0.9517 | 0.9526 | 0.9573 | **0.9604** |
| Pima diabetes | *Smith et al. (1988)* | 0.7328 | 0.7262 | 0.7095 | **0.7331** |
| Lung cancer | *Hong & Yang (1991)* | 0.7083 | 0.6042 | 0.6927 | **0.8021** |
| Cardiotocography | *Ayres-de Campos et al. (2000)* | 0.9948 | 0.9948 | 0.9925 | **0.9958** |
| SPECTF heart | *Kurgan et al. (2001)* | 0.9470 | **0.9492** | 0.9462 | 0.9463 |
| Arcene | *Guyon et al. (2005)* | 0.8108 | 0.8433 | **0.8900** | 0.8455 |
| Colposcopy QA | *Fernandes, Cardoso & Fernandes (2017b)* | 0.7760 | 0.8122 | 0.7961 | **0.8470** |
| **Best** | | 1 | 2 | 1 | 5 |

**Note:**
We highlight the best performing models in bold.

**Table 6 Performance of the proposed architectures on other datasets downloaded from UC Irvine Machine Learning Repository (*University of California Irvine, 1987*), measured through logarithmic loss.**

| Dataset | | Baseline | Semi | Sym | Zero |
|---|---|---|---|---|---|
| Breast cancer | *Mangasarian, Street & Wolberg (1995)* | 0.0984 | **0.0888** | 0.0966 | 0.0930 |
| Mammographic | *Elter, Schulz-Wendtland & Wittenberg (2007)* | 0.5122 | 0.5051 | 0.4973 | **0.4822** |
| Parkinson | *Little et al. (2007)* | 0.3945 | 0.4042 | **0.3883** | 0.4323 |
| Pima diabetes | *Smith et al. (1988)* | 0.5269 | **0.5229** | 0.5250 | 0.5472 |
| Lung cancer | *Hong & Yang (1991)* | 1.1083 | 0.8017 | **0.6050** | 0.8328 |
| Cardiotocography | *Ayres-de Campos et al. (2000)* | 0.0113 | 0.0118 | 0.0116 | **0.0110** |
| SPECTF heart | *Kurgan et al. (2001)* | **0.4107** | 0.4205 | 0.4121 | 0.4196 |
| Arcene | *Guyon et al. (2005)* | 1.3516 | **0.8855** | 1.0230 | 1.1518 |
| Colposcopy QA | *Fernandes, Cardoso & Fernandes (2017b)* | 0.5429 | 0.5406 | 0.5195 | **0.4850** |
| **Best** | | 1 | 3 | 2 | 3 |

**Note:**
We highlight the best performing models in bold.

For visualization purposes, we are using t-SNE based upon neighborhood similarities, since learning a valuable representation in a two-dimensional space raises difficulties. Moreover, because of the high dimensionality of our embeddings, their reduction capabilities rely on their sparsity.

## Results in other applications

To observe the impact of our method, we validated the performance of the aforementioned model architectures on several biomedical datasets available on the UC Irvine Machine Learning Repository. Thus, we assessed the model's performance on nine datasets. The machine learning models we proposed achieved high prediction results, being the zero-mapping approach the best model in most cases (Tables 5 and 6).

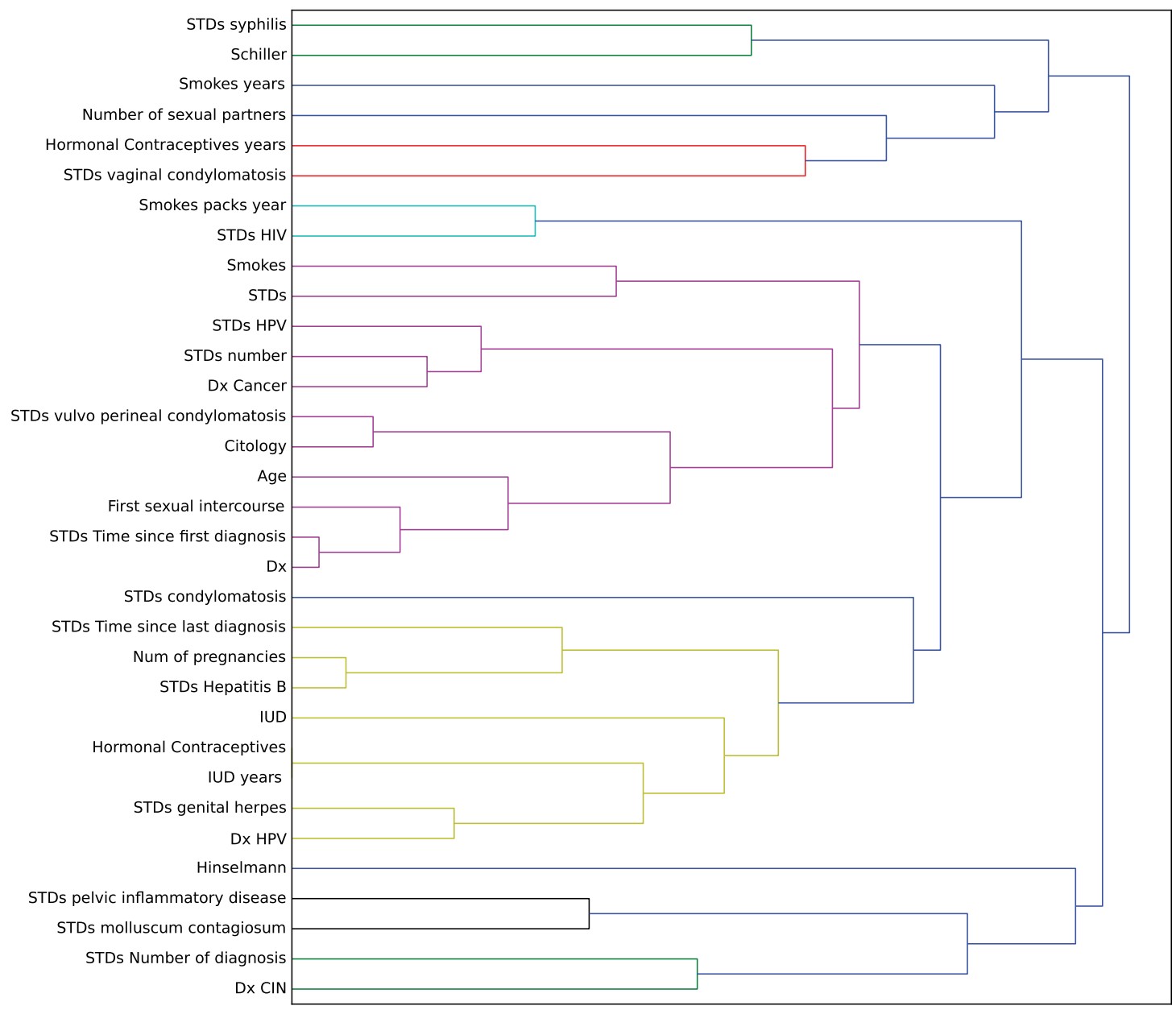

**Figure 7  Agglomerative clustering of features by impact on the embedding space.**

This outcome suggests that mapping the majority class to a unique point in the space might improve the learning effectiveness in unbalanced settings. This idea draws a link between binary and one-class classification, and we plan to explore it more in the future.

## DISCUSSION

As shown in the Results section, our deep learning algorithm can predict cervical cancer diagnosis with high accuracy. To further understand the clinical interpretability of our prediction model, we investigated which dataset risk features have the highest impact in the cervical cancer diagnosis for the patients.

In fact, pre-invasive intra-epithelial lesions of the cervix and cervical cancer relate to HPV infection of oncological serotypes that progress to oncological lesions, and multiple factors contribute to this progress without a definite cause-dependent relation. The patterns that have highest acceptance in the literature regard presence of human immunodeficiency virus and smoking, followed by sexual risk behaviors such as early sexual initiation, promiscuity, multiple pregnancies, and a history of sexually transmitted infections. Another factor involved is the use of oral contraceptives.

From a technical point of view, while black-box machine learning techniques have achieved state-of-the-art results in several applications, the lack of interpretability of the induced models can limit their general acceptance by the medical community. Thus, we tried to understand the relationships by using our prediction model to corroborate if they are supported by the medical literature.

In this context, we studied the impact of the original features on the embedding space to find correlations in the decision process. To determine this impact, we perturbed each feature using all the other values from the feature's domain, and then we computed the maximum impact of the features in the embedded space. Finally, we applied an agglomerative clustering technique to aggregate features with similar impact in the embedding features. From a medical point of view, we validated several properties of interest (Fig. 7).

For instance, risky sexual patterns such as an early sexual initiation and the presence (and lifespan) of STDs (with a special focus on HPV) have the most similar impact in the predictive outcome of the model. Also, smoking habits are associated by the model as having a similar effect as these sexual patterns. These relationships were already studied in the medical literature (*Louie et al., 2009*; *Deacon et al., 2000*).

The similarity between the use of hormonal contraceptives with condylomatosis and the use of intrauterine devices with STDs shows another interesting pattern that has not been quantified yet to the best of our knowledge. These patterns might be evidence of sexual patterns with high risk.

## CONCLUSION

Cervical cancer is still a widespread disease nowadays, and its diagnosis often requires frequent and very time-consuming clinical exams. In this context, machine learning can provide effective tools to speed up the diagnosis process, by processing high-scale patients' datasets in a few minutes.

In this manuscript, we presented a computational system for the prediction of cervical patient diagnosis, and for the interpretation of its results. Our system consists of a loss function that allows joint optimization of dimensionality reduction, and classification techniques able to promote relevant properties in the embedded spaces. Our deep learning methods predicted the diagnosis of the patients with high accuracy, and their application to other datasets showed that their robustness and effectiveness is not bounded to cervical cancer. Our methods can be used to analyze profiles of patients where the biopsy and potentially other screening results are missing, and are able to predict confidently if they have cervical cancer.

In the future, we plan to employ alternative approaches for data missing imputation, such as oversampling through *k*-nearest neighbors (*Santos et al., 2015*) or latent semantic indexing similarity (*Chicco & Masseroli, 2015*). We also plan to try alternative prediction models, like probabilistic latent semantic analysis (*Pinoli, Chicco & Masseroli, 2015*). Finally, we plan to extend our computational system by adding a feature selection step, able to state the most relevant features among the dataset.

## ACKNOWLEDGEMENTS

The authors thank the Gynecology Service of the Hospital Universitario de Caracas, and Francis Nguyen (Princess Margaret Cancer Centre) for the English proof-reading of this manuscript.

### Funding

This work was funded by the Project "NanoSTIMA: Macro-to-Nano Human Sensing: Towards Integrated Multimodal Health Monitoring and Analytics/NORTE-01-0145-FEDER-000016" financed by the North Portugal Regional Operational Programme (NORTE 2020), under the PORTUGAL 2020 Partnership Agreement, and through the European Regional Development Fund (ERDF), and also by Fundacao para a Ciencia e a Tecnologia (FCT) within the PhD grant number SFRH/BD/93012/2013. The funders had no role in study design, data collection and analysis, decision to publish, or preparation of the manuscript.

### Grant Disclosures

The following grant information was disclosed by the authors:
NanoSTIMA: Macro-to-Nano Human Sensing: Towards Integrated Multimodal Health Monitoring and Analytics: NORTE-01-0145-FEDER-000016.
North Portugal Regional Operational Programme: NORTE 2020.
PORTUGAL 2020 Partnership Agreement.
European Regional Development Fund (ERDF).
Fundacao para a Ciencia e a Tecnologia (FCT): SFRH/BD/93012/2013.

### Competing Interests

The authors declare that they have no competing interests.

### Author Contributions

- Kelwin Fernandes conceived and designed the experiments, performed the experiments, analyzed the data, contributed reagents/materials/analysis tools, prepared figures and/or tables, performed the computation work, authored or reviewed drafts of the paper, approved the final draft.
- Davide Chicco conceived and designed the experiments, analyzed the data, contributed reagents/materials/analysis tools, prepared figures and/or tables, authored or reviewed drafts of the paper, approved the final draft.

- Jaime S. Cardoso conceived and designed the experiments, contributed reagents/ materials/analysis tools, authored or reviewed drafts of the paper, approved the final draft, proposed parts of the general strategy of the project.
- Jessica Fernandes analyzed the data, contributed reagents/materials/analysis tools, authored or reviewed drafts of the paper, approved the final draft, construction of the dataset and domain expertise about the application.

### Data Availability

The dataset is publicly available at the University of California, Irvine Machine Learning Repository: https://archive.ics.uci.edu/ml/datasets/Cervical+cancer+%28Risk +Factors%29

The dataset is also available at Github: https://github.com/kelwinfc/cervical-cancer-screening/tree/master/risk-factors/data

The software code of the methods used in the project is available at Github: https://github.com/kelwinfc/cervical-cancer-screening/

We implemented the software in Python 2.7 using the Keras (*Chollet, 2015*) and TensorFlow (*Abadi et al., 2016*) frameworks, and tested it on a computer running the Linux Ubuntu 16.04 operating system.

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
