# Peer review of "Supervised deep learning embeddings for the prediction of cervical cancer diagnosis"

_PeerJ Computer Science, doi:10.7717/peerj-cs.154_

## Round 0.1 · original submission · Major Revisions

Both reviewers consider the paper is interesting and deserve to be published, but they also consider there are some issues that needs review before this acceptation. Try to address all of them in a next version of the paper.

Reviewer 1 ·

Basic reporting

Section Introduction

-“The two most widespread techniques in screening programs are cytology, either conventional or liquid,
38 and colposcopy [11, 26, 12, 35].” -> This statement appears to be unrelated to the place where it is located in the introduction.

- “We applied the our learning scheme…” -> check this.

Section Methods

- “In many cases, researchers perform these two steps separately, deriving suboptimal combined models.”-> citation needed

- The description of the proposed method should be improved.

- It is not clear which ideas portrayed are original or whether they have been proposed before in other works.

Section Dataset

-“The hospital anonymized all the records before releasing the dataset The dataset is” -> check this

-“normalization, and we inputed missing data using the average value [4].” -> Why authors used this method to the estimation of missing values?. There are several methods more accurate than the used one.

Section Software and Data Availability

- The URL https://github.com/kelwinfc/cervical-cancer-screening is not working.
- The URL https://archive.ics.uci.edu/ml points to the home of UCI repository and not to the dataset used.

Experimental design

- A better explanation of the steps carried out to tune the neural network is required.
- Some parameters configurations taken by the authors are not clear and they require a better explanation. For instance, why authors used RMSProp and no other optimization algorithm.?
Section Results in other applications

- More than six biomedical datasets should be considered to demonstrate the effectiveness of the method. Also, the results must be analysed by means of non-parametric test to detect significant differences between the configurations.
- On the other hand, it would be interesting if the authors compare their method to other well-established classification algorithms.

Validity of the findings

Section Discussion

“From a technical point of view, while machine learning techniques have achieved state-of-the-art
results in several applications, the lack of interpretability of the induced models can limit their general
acceptance by the medical community.” -> This statement is partially wrong, due to there are several machine learning approaches that return interpretable models such as rule-based methods and decision tree-based methods. Even, there are several works proposing the extraction of rules from the results of black-bocks-based models.

- The discussion and conclusions are well stated, they are limited to the results obtained.

Additional comments

I consider that this work is interesting, but it should be significantly improved before it could be considered for publication. I suggest that authors review the work attending the comments made by the reviewer.

Reviewer 2 ·

Basic reporting

The article is well written but there are some typos:

line 67: please replace "we applied the our..." with "we applied our..."
line 76: please replace "data high dimensionality" with "high dimensionality data"
line 144: please replace "autoencoders can to learn" with "autoencoders can learn"
line 174: please replace "behavior related different" with "behavior related to different"

Experimental design

Methods Section of the paper needs some improvements.
As for the dataset, what are the type and numebr of features? Apart from the reference [32], please add a brief description of the dataset.
The paper lacks of some technical details.
First of all, how did you choose the number of features for the dimensionality reduction phase? please clarify
The set of parameters (Table 1) of neural network should be better described.
Please add some implementation details such as:
- programming language
- adopted deep learning framework
- execution times
On line 198, for the first time you speak about a semi-supervided approach: please clarify
On Table 2 and 3, the first row to which subese is referred to?

Validity of the findings

What is the performance gain provided by the dimensionality reduction procedure? please clarify
In order to make the paper more interesting, you should compare your ANN classifier with other well known classification algorithms, such as SVM, Naive Bayes, k-NN, random forest, using both the original dataset and the low dimensional one.

---

## Round 0.2 · Minor Revisions

Please follow the recommendations of reviewer 1, mainly in what corresponds to performing a statistical treatment of results.

Reviewer 1 ·

Basic reporting

Section Introduction

“However, many projects that take advantage of dimensionality reduction and classification use suboptimal approaches, where each component is learned separately.” → citation needed.

Section Methods

“We use a Parametric Rectifier Linear Unit (PReLU) [20] as activation function in the hidden layers of our architectures (Figure 1).” → Why the authors use PreLU?, it may be ok, but some justification must be given in this regard.

Section Dataset

- I do’nt understand very well what means bool × int, int x int, etc.

Experimental design

- Statistical comparisons by means of non-parametric test should be carried out in order to draw conclusions more reliable. For example, what is the model that, in average, obtains the more significant results in Table 6?

Validity of the findings

- The discussion and conclusions are well stated, they are limited to the results obtained.

Additional comments

I consider that the work was improved; the authors addressed most of the comments raised in the previous version.

Reviewer 2 ·

Basic reporting

no comment

Experimental design

no comment

Validity of the findings

no comment

Additional comments

The authors succesfully answered all my remarks, therefore their paper can now be accepted

---

## Round 0.3 · accepted · Accept

I think that you have followed all the recommendations of the reviewers, and paper is now ready for publication.

#